# Long COVID as a Tauopathy: Of “Brain Fog” and “Fusogen Storms”

**DOI:** 10.3390/ijms241612648

**Published:** 2023-08-10

**Authors:** Adonis Sfera, Leah Rahman, Carlos Manuel Zapata-Martín del Campo, Zisis Kozlakidis

**Affiliations:** 1Paton State Hospital, 3102 Highland Ave, Patton, CA 92369, USA; 2School of Behavioral Health, Loma Linda University, 11139 Anderson St., Loma Linda, CA 92350, USA; 3Department of Psychiatry, University of California, Riverside 900 University Ave, Riverside, CA 92521, USA; 4Department of Neuroscience, University of Oregon, 222 Huestis Hall, Eugene, OR 97401, USA; rahman.leah@dsh.ca.gov; 5Instituto National de Cardiologia, Juan Badiano 1, Belisario Domínguez Secc 16, Tlalpan, Ciudad de México 14080, Mexico; carloszapatamc@gmail.com; 6International Agency for Research on Cancer, World Health Organization, 69000 Lyon, France; kozlakidisz@iarc.who.int

**Keywords:** tauopathy, phosphorylated tau, cell–cell fusion, cell senescence, taVNS

## Abstract

Long COVID, also called post-acute sequelae of SARS-CoV-2, is characterized by a multitude of lingering symptoms, including impaired cognition, that can last for many months. This symptom, often called “brain fog”, affects the life quality of numerous individuals, increasing medical complications as well as healthcare expenditures. The etiopathogenesis of SARS-CoV-2-induced cognitive deficit is unclear, but the most likely cause is chronic inflammation maintained by a viral remnant thriving in select body reservoirs. These viral sanctuaries are likely comprised of fused, senescent cells, including microglia and astrocytes, that the pathogen can convert into neurotoxic phenotypes. Moreover, as the enteric nervous system contains neurons and glia, the virus likely lingers in the gastrointestinal tract as well, accounting for the intestinal symptoms of long COVID. Fusogens are proteins that can overcome the repulsive forces between cell membranes, allowing the virus to coalesce with host cells and enter the cytoplasm. In the intracellular compartment, the pathogen hijacks the actin cytoskeleton, fusing host cells with each other and engendering pathological syncytia. Cell–cell fusion enables the virus to infect the healthy neighboring cells. We surmise that syncytia formation drives cognitive impairment by facilitating the “seeding” of hyperphosphorylated Tau, documented in COVID-19. In our previous work, we hypothesized that the SARS-CoV-2 virus induces premature endothelial senescence, increasing the permeability of the intestinal and blood–brain barrier. This enables the migration of gastrointestinal tract microbes and/or their components into the host circulation, eventually reaching the brain where they may induce cognitive dysfunction. For example, translocated lipopolysaccharides or microbial DNA can induce Tau hyperphosphorylation, likely accounting for memory problems. In this perspective article, we examine the pathogenetic mechanisms and potential biomarkers of long COVID, including microbial cell-free DNA, interleukin 22, and phosphorylated Tau, as well as the beneficial effect of transcutaneous vagal nerve stimulation.

## 1. Introduction

The COVID-19 pandemic may be succeeded by prolonged sequelae that could affect the life quality of many patients while at the same time increasing medical complications and healthcare expenditures. Cognitive impairment, one of the lingering symptoms, often referred to by patients as “brain fog”, remains poorly defined; however, viral presence in reservoirs and the “seeding” of pathological Tau may account for cognitive difficulties.

Tauopathies are neurodegenerative disorders marked by the accumulation of neurofibrillary tangles and paired helical filaments in neurons and glia, leading to apoptosis. Tauopathies can be primary (Tau is the main disease driver) or secondary (Tau aggregation is triggered by a different insult). The most common secondary tauopathies—frontotemporal dementia (FTD) and Alzheimer’s disease (AD)—comprise the largest share of late-life neurodegenerative disorders, linking pTau to this pathology [1]. Tau is encoded by the microtubule-associated protein tau (MAPT) gene and has been implicated in the homeostasis of axonal and synaptic transmission [2]. Several tauopathies, including subacute sclerosing panencephalitis and postencephalitic parkinsonism, have been associated with viral infections, suggesting that long COVID may be driven by a similar mechanism [3,4,5].

Long COVID refers to the clinical manifestations that develop during or after COVID-19 infection and continue for at least 12 weeks [6]. Due to the absence of specific tests for this syndrome, long COVID is currently diagnosed by history taking and physical examination. For this reason, it is difficult to differentiate long COVID from severe acute respiratory syndrome coronavirus 2 (SARS-CoV-2) reinfection or the activation of dormant viruses, such as Herpes virus (HSV) [7]. By the same token, COVID-associated neurocognitive dysfunction (CAND) cannot be readily parsed out from other cognitive dysfunctions, whether new or pre-existing [8].

To effectively infect an organism, enveloped viruses must first usurp the plasma membrane fusogens and coalesce with host cells. Upon successful entry into the cytosol, the SARS-CoV-2 virus proceeds to fuse host cells with each other, forming large, multinucleated syncytial structures. These formations enhance infectivity by allowing infection as well as hyperphosphorylated Tau (pTau—a newly identified marker of long COVID—to spread “from within”, increasing the efficacy of dissemination. For example, the fusion of infected with noninfected cells proliferates the contagion more rapidly than invading cells one by one [9,10,11,12]. Furthermore, lipopolysaccharide (LPS) was demonstrated to enhance the detrimental effects of pTau, linking this gut-originating endotoxin to tauopathies [13].

In a previous article, we connected the SARS-CoV-2 virus with premature endothelial senescence, gut-barrier dysfunction, and the subsequent translocation of microbes and/or their molecules into host tissues and organs, including the brain [14]. Gut microbes are immunologically tolerated in the gastrointestinal (GI) tract, however, depending on the microorganism, they may trigger vehement immunogenicity upon migration across the lamina propria [15]. Indeed, novel studies have identified microbial DNA, and LPS in the peripheral blood of COVID-19 patients, suggesting that this virus facilitates translocation [16,17,18,19].

Novel studies have shown that the S protein of the SARS-CoV-2 virus binds directly to LPS, facilitating contagion as well as inflammation [20,21]. In addition, the S antigen-associated fusogen—Ca^2+^/calmodulin-dependent protein kinase II (CaMKII)—implicated in cognitive dysfunction promotes cell–cell fusion, augmenting the virulence of SARS-CoV-2 [22,23]. Moreover, viral attachment to angiotensin-converting enzyme-2 (ACE-2), upregulates angiotensin II (ANG II), contributing to both Tau hyperphosphorylation and dysfunctional acetylcholine (ACh) signaling [14,24,25] (Figure 1).

Mitochondria, crucial for host innate immunity, are exploited by many viruses, including SARS-CoV-2, to avert detection and ensure proper replication in host cells. For example, SARS-CoV-2 antigens (nonstructural protein 4 and 9 (NSP4, NSP9) and open reading frame 9C (ORF9C) disrupt the organelle, generating mitochondrial reactive oxygen species (mROS), which promote the development of pTau [9,26,27,28]. In this regard, assays of phosphorylated Tau at threonine 217 (pTau-217) and threonine 181 (pTau-181), developed by Lilly and Simoa^®®^, are being utilized as tauopathy blood markers, respectively. We suggest that the presence of these biomolecules can also accurately diagnose long COVID (Table 1).

Novel studies have shown that premature cellular senescence promotes pTau and upregulates intracellular calcium (Ca^2+^), activating the fusion machinery [29,30,31,32] (Figure 2). For example, Ca^2+^, released from the endoplasmic reticulum (ER) and/or imported from the extracellular compartment activates TMEM16F, a lipid scramblase that triggers cell–cell fusion by externalizing phosphatidylserine (ePS) [33] (Figure 2). In addition, premature cellular senescence upregulates intracellular iron, a biometal indispensable for viral proliferation that may increase oxidative stress and telomere erosion, causing further senescence.

The infection with the SARS-CoV-2 virus has been shown to activate human endogenous retroviruses (HERVs), a large source of ancestral fusogens likely implicated in long COVID [34,35]. HERVs are embedded in the human DNA and, with few exceptions, are epigenetically suppressed. However, under pathological circumstances, these viral fossils can be expressed, causing pathology, including cancer and neurodegenerative disorders [36].

In this perspective article, we take a closer look at long COVID as a probable tauopathy, discussing potential biomarkers such as microbial cell-free DNA (mcfDNA), interleukin 22 (IL-22), and phosphorylated Tau (pTau).

We also review the beneficial effects of noninvasive vagus nerve stimulation as a therapeutic intervention for long COVID.

**Table 1 ijms-24-12648-t001:** Potential markers for microbial translocation and tauopathy.

Marker Type	Marker	Assay	References
Gut barrier	IL-22	Singulex-Erenna^®®^	[37]
Gut barrier	mcfDNA	Karius Test^®®^	[38]
Neurodegeneration	pTau 217	Lilly	[39]
Neurodegeneration	pTau 181	Simoa^®®^	[40]
NLR	CIC	Entotic cell death assays	[41]

Example: downregulated IL-22 with upregulated mcfDN, pTau, and NLR would reflect microbial translocation and the emergence of tauopathy. (Table 1).

## 2. SARS-CoV-2 Reservoirs

The symptoms of long COVID are likely caused by a viral remnant in select body reservoirs where the pathogen is shielded from host defenses. For example, at 15 months post-infection follow-up, the S protein of the SARS-CoV-2 virus was found in monocytes, indicating that these cells could comprise viral reservoirs [42]. Other studies have found that intestinal epithelial cells (IECs), adipocytes, and endothelial cells (ECs), as well as astrocytes and microglia, could serve as potential SARS-CoV-2 reservoirs [43,44,45,46]. In sanctuary cells, the virus has been found to associate with ferritin and pTau, linking this pathogen to protein misfolding and iron dyshomeostasis [47,48]. Interestingly, monocytes, microglia, and macrophages are known human immunodeficiency virus (HIV) reservoirs, suggesting that they could also harbor the SARS-CoV-2 virus [49].

At the molecular level, the AhR/STAT3/IL-22 axis exhibits antiviral and anti-inflammatory properties as it connects the cholinergic anti-inflammatory pathway (CAP) with gut microbes. In this regard, transcutaneous auricular vagus nerve stimulation (taVNS), a modality that increases the antiviral neurotransmitter ACh, may improve both cognition and the gut-barrier function [50,51] (Figure 1). Conversely, the aberrant phosphorylation of the AhR/STAT3/IL-22 axis can exacerbate viral infection and microbial translocation [52,53].

In the CNS, long COVID was associated with gray matter loss, likely accounting for the impaired attention, concentration, and executive function documented in this condition [44,54,55,56]. Several studies have linked long COVID to the reactive astrocytes, which the virus may enter via a basigin (BSG) receptor, a protein implicated in microvascular disease [57,58]. This is significant as astrocytes, established HIV reservoirs, are essential for the neuro-vascular unit and may easily capture the SARS-CoV-2 virus from the cerebral circulation [59,60,61]. In addition, as microglia are known HIV reservoirs, these cells may promote long COVID by also housing the SARS-CoV-2 virus [62]. Indeed, both astrocytes and microglia can assume neurotoxic phenotypes, eliminating healthy neurons and synapses—a process that could account for “brain fog” [44].

The likely existence of SARS-CoV-2 viral reservoirs highlights the unmet need of eradicating this as well as the HIV virus from the sanctuary cells. This brings CAP to the forefront of long COVID treatment. Indeed, at present, no α7nAChR agonist has been approved (and nicotine is a nonspecific contributor), and VN stimulation (VNS) may be the only available modality for CAP augmentation and the removal of viruses from reservoirs [63,64]. The activation of α7nAChR in persons living with HIV has ameliorated HIV-associated neurocognitive disorders (HAND), suggesting that this modality can clear viral reservoirs, areas where the highly active antiretroviral therapy (HAART) cannot access [65]. We hypothesize that taVNS can clear both SARS-CoV-2 and HIV from viral reservoirs by activating AhR/STAT3/IL-22.

## 3. SARS-CoV-2 Virus-Induced Cell–Cell Fusion

Enveloped viruses, including SARS-CoV-2, are known for inducing premature cellular senescence, a program of permanent replication arrest with an active metabolism, generating a unique secretome known as senescence-associated secretory phenotype (SASP). They upregulate intracellular iron, and Ca^2+,^ senescent and fused cells are hospitable to viruses that require these elements to replicate [66,67,68]. Fused cells, characterized by ePS, promote immunosuppression, enabling the virus to avert host immunity [69]. However, cellular senescence-upregulated intracellular iron may trigger ferroptosis, a nonapoptotic cell death caused by the oxidation of endoplasmic reticulum (ER) lipids in the absence of antioxidants. Indeed, due to SARS-CoV-2’s affinity for astrocytes, the virus likely disrupts the generation of glutathione peroxidase 4 (GPX4), a selenocysteine, which is in charge of repairing oxidized phospholipids. GPX4 is synthesized by astrocytes and shuttled to the neurons, where it prevents lipid peroxidation and ferroptosis [70].

In contrast to ferroptosis, which involves neuronal death, ferrosenescence is a phenotype in which the cells remain alive but iron-damaged genomes enable the mobilization of transposable elements (TEs) known colloquially as” jumping genes”. TEs are DNA segments that can extricate themselves from the genome and reinsert in the double helix at a different location. TE mobilization has been associated with “fusogen storms”—the excessive release of fusion molecules from HERVs (viruses acting as TEs) [71]. Moreover, pTau was demonstrated to further mobilize TEs, enhancing fusogen storms while generating a hospitable milieu for viral progeny [72].

It is unclear at this time whether fused neurons are viable in humans and for what length of time. A recent study in *Caenorhabditis elegans* (*C. elegans*) found that an aberrant expression of fusogens in neurons alters the connectome as well as the behavior, although sensation remains intact [25]. It is difficult to extrapolate these data to humans; however, connectivity was demonstrated to affect cognition [73].

In a previous article, we defined ferrosenescence as “neurodegeneration-by-iron”, that is, premature molecular aging due to iron-induced damage to both DNA and the genomic repair systems, especially the p53 [74]. Since p53 also drives natural killer cells (NKCs), which under normal circumstances remove ferrosenescent cells, dysfunctional p53 may lead to iron deposition in the central nervous system (CNS) [75]. Indeed, excessive iron in the caudate and putamen was documented in long COVID, linking this condition to ferrosenescence [76,77].

The SARS-CoV-2 virus can alter iron metabolism by several mechanisms. For example, the S1 protein/ACE-2 attachment upregulates ANG II, increasing intracellular iron via angiotensin II type 1 receptors (AT-1Rs) (Figure 3). The S2 antigen of the SARS-CoV-2 virus disables p53, inducing ferrosenescence due to unrepaired DNA. The virus also attacks hemoglobin via ORF3 and ORF10 antigens, releasing more iron for viral replication [78,79]. On the other hand, sartans and iron chelators avert ferrosenescence by blocking AT-1Rs and lowering the excess iron (Figure 3).

Both physiological and pathological aging were associated with syncytia formation and excessive intracellular Ca^2+^ and iron [80,81]. Moreover, the SARS-CoV-2 virus disseminates pTau to healthy cells, both directly by piercing the cell membranes and indirectly via extracellular vesicles (EVs) [82,83].

## 4. Fusion of Postmitotic Cells: Heart and Brain Syncytia

Several body tissues are structured as anatomical or functional syncytia. For example, the cardiac muscle maintains distinct cellular boundaries while functioning as a syncytium [84]. In contrast, skeletal muscle fibers and placental villous trophoblasts form anatomical syncytia, generating giant structures with a common cytoplasm and multiple nuclei [85].

Several viruses are known for inducing pathological cell–cell fusion, which in the heart and skeletal muscle may be superimposed on physiological syncytia, causing pathology. Viral pathogens can also drive postmitotic cells such as cardiomyocytes and myocytes to re-enter mitosis, probably contributing to cardiomyopathy in the myocardium and fatigue, frailty, or weakness in skeletal muscle [86,87].

In the CNS, astrocytes and oligodendrocytes form syncytia as they communicate via connexin channels [88]. These structures may arise from postmitotic neurons re-entering the cell cycle and forming aneuploid or polyploid cells, which is a phenomenon reported more than two decades ago [29]. As opposed to the early studies that assumed that fused neurons always undergo apoptosis, new research has found that these cells can stay indefinitely in fused states and remain functional [89,90].

The idea of cell–cell fusion in the brain has been around since the 19th century when Golgi and Cajal had opposite views on whether neurons were distinct entities or formed syncytial structures [91]. The advent of electron microscopy in the middle of the 20th century revealed that the neurons were individual cells, while astrocytes formed functional syncytia [92].

Under pathological circumstances, including viral infections, neurons can fuse with each other or with glial cells, forming syncytia (Figure 4). For example, the fusion of Purkinje neurons with bone marrow cells was demonstrated in both humans and rodents, suggesting that neurons can fuse not only with each other but also with somatic cells [29,93]. In addition, multinucleated neurons in the supraoptic nucleus were found in patients with pneumonia, suggesting that fusogenic viruses can access the brain [94]. Moreover, neuronal syncytia were demonstrated in both normal and pathological aging, suggesting that aneuploidy and polyploidy could be more common in the human brain than currently believed [95,96]. Indeed, somatic mosaicism (genomic differences from neuron to neuron) was discovered with the help of a new technique capable of discerning chromosomes in individual cells [97]. The fact that neurons have different DNA from one cell to another is significant and may be the result of genome reorganization after viral infections or TE mobilization. This suggests that viruses may contribute to the variability of human brain function in both health and disease.

Taken together, cellular senescence and cell–cell fusion are physiological processes exploited by viruses to facilitate replication in human cells. Upregulated iron and Ca^2+^ in senescent cells generate a virus-friendly milieu while increasing the risk of host cognitive dysfunction.

## 5. SARS-CoV-2 Virus Activates Human Endogenous Retroviruses (HERVs)

Human endogenous retroviruses (HERVs) are viral relics that were integrated into the genome in ancient times and currently comprise about 8% of human DNA [98]. Most HERVs consist of damaged genes that cannot be transcribed and are suppressed by p53, which, aside from repairing the genome, blocks viruses from integrating into human DNA [99]. To overcome p53, cancers and viruses often exploit ubiquitination, an epigenetic mechanism that terminates the action of p53 [100].

Cell–cell fusion, a physiological mechanism of wound healing, is activated by cell membrane lesions, which drive the “wounded” cell to fuse with its healthy neighbors, preventing apoptosis by spillage of the intracellular content [101]. Viruses exploit wound healing by puncturing cell membranes to activate the fusion machinery [102]. To accomplish this task, the SARS-CoV-2 virus exploits arginine, a unique amino with side chains capable of piercing cell membranes [103,104]. For example, the S antigen of the SARS-CoV-2 virus contains a polybasic cleavage motif, “Proline-Arginine-Arginine-Alanine (PRRA)”, in which the bi-arginine forms a pore in the cell membrane, initiating fusion [105]. Aside from utilizing arginine, the SARS-CoV-2 virus also activates HERVs, providing additional fusogens—a “plan B”—for ensuring viral entry [34].

Over thousands of years, several HERVs have been “domesticated” and have assumed physiological functions in the human body. For example, HERV-W ENV encodes for a physiological placental fusogen, syncytin-1, which plays a key role in pregnancy by enabling the formation of syncytiotrophoblasts [106]. In addition, HERV-FRD, another “domesticated” endogenous retrovirus, encodes for syncytin-2, a fusogen that also plays a key role in placentation [107,108]. In the CNS, the neuronal gene Arc is a “domesticated” HERV and a master regulator of synaptic plasticity; it is derived from the ancestral Ty3/gypsy retrovirus, implicated in memory formation [109].

DNA damage can awaken suppressed HERVs, triggering pathology, including cancer and neuropsychiatric illness [110]. As p53 opposes the transcription of HERVs as well as the integration of exogenous viruses into human DNA, SARS-CoV-2 can release fusogens by blocking p53, disinhibiting the transcription of syncytin-1 and 2 [111] (Figure 5). In consequence, enhanced syncytia formation increases the systemic symptoms of long COVID, including “brain fog”, by several mechanisms, including HERV activation.

In the brain, excessive syncytin-1 can activate several pro-inflammatory and autoimmune cascades, triggering neuropsychiatric pathology, including multiple sclerosis (MS) [112]. Moreover, the SARS-CoV-2 inhibition of p53 destabilizes the genome, mobilizing TEs and facilitating the transcription of fusogens, including syncytins [113]. Along this line, impaired p53 predisposes to cancer, suggesting that SARS-CoV-2 may be an oncogenic virus [114,115,116]. Indeed, pTau was implicated in tumorigenesis, while HERVs spread pTau, connecting ancient viruses with present-day cancer [82]. In addition, since COVID-19 disrupts DNA repair, it may inhibit cell-cycle arrest genes, inducing postmitotic cells to engage in mitosis [117].

## 6. Microbial Translocation

It has been established that the SARS-CoV-2 virus employs several mechanisms to induce premature endothelial senescence and a dysfunctional intestinal and blood–brain barrier (BBB). This contributes to the translocation of microbes and/or their molecules from the GI tract into host tissues and organs, eventually reaching the brain (Figure 6).

Microbial translocation from the GI tract into the systemic circulation attracted the attention of clinicians and researchers during the human immunodeficiency virus 1 (HIV-1) epidemic in the 1980s. This virus has been known for depleting gut interleukin 22 (IL-22), leading to increased intestinal permeability and microbial dissemination in host tissues [118].

Various microbial translocation markers have been described since the 1980s, including LPS-binding protein (LBP), soluble CD14 (sCD14), and endotoxincore antibody (EndoCAb) [119]. We propose that IL-22 and microbial cell-free DNA (mcfDNA) comprise second-generation, high-quality markers of intestinal permeability and microbial migration. In addition, we believe that pTau 217 and pTau 181 can identify tauopathies as well as long COVID.

Neutrophil-to-lymphocyte ratio (NLR) is an inflammatory marker that may be elevated in long COVID due to lymphopenia. Low lymphocyte count may be caused by cell-in-cell (CIC) structures, which are phenomena described by Karl J. Eberth in the 19th century. CIC formations refer to cells internalized by other cells and are phenomena observed in cancer and lymphocytes-in-IECs documented in both COVID-19 and long COVID [120,121]. This may be significant as the internalized lymphocytes, likely infected with SARS-CoV-2, may comprise unique viral reservoirs, maintaining long COVID.

### 6.1. Interleukin 22 (IL-22)

Discovered in 2000, IL-22 is a member of the IL-10 family of cytokines generated by several lymphocyte types, including T helper (Th) 17 cells, γδ T cells, NKCs, and innate lymphoid cells (ILCs) [122]. IL-22 controls several IEC functions, including mucus formation, permeability, and the synthesis of complement and antimicrobial peptides (AMPs), indicating that this cytokine functions as the master regulator of gut-barrier permeability [123].

The crosstalk between IL-22 and its receptor (IL-22R), a dimeric protein comprised of IL-22R1 and IL-10R2, activates the JAK/STAT pathway, a key antibacterial and antiviral system that protects against gut pathogens [124,125]. IL-22 exerts antiviral properties against COVID-19 and influenza, as well as beneficial action in inflammatory bowel disease (IBD) [126,127,128,129]. Singulex-Erenna^®®^, a standardized IL-22 peripheral blood assay, should be studied further not only for IBD and viral infections but also for neuropsychiatric pathology [37].

### 6.2. Microbial Cell-Free DNA (mcfDNA)

mcfDNA is a noninvasive, high-throughput sequencing technology that detects pathogens in the peripheral blood [130]. Presently, mcfDNA has only been utilized in sepsis; however, given its high sensitivity and specificity, it likely comprises a valid biomarker of gut permeability [131]. For example, the Karius Test^®®^, developed in 2017, can detect mcfDNA in the peripheral blood, suggesting that it could be a diagnostic tool for the microorganisms associated with neuropsychiatric disorders [38].

### 6.3. Hyperphosphorylated Tau (pTau)

pTau, a central marker of tauopathy, has been detected in several peripheral organs, including the gut, in which it is associated with IBD [132,133]. Aside from the aggregated long fibrils, several studies found short pTau in COVID-19-associated peripheral neuropathy [134]. pTau 217 and pTau 181 have been patented by Lilly and Simoa^®®^ for AD; however, they could also diagnose long COVID [39,40].

### 6.4. Neutrophil-to-Lymphocyte Ratio (NLR)

NLR is obtained from the peripheral blood and reflects the ratio between innate immunity (neutrophils) and the adaptive immune system (lymphocytes) [135]. Low lymphocyte counts, lasting for up to 16 weeks after the acute COVID-19 illness were detected in many patients, suggesting that lymphopenia could be a reliable biomarker of long COVID [136].

Lymphopenia may be caused by cell-in-cell (CIC) structures, which are phenomena referring to lymphocytes internalized in IECs [41,120,137]. CIC structures, commonly found in cancer, can be driven by mitosis and potentially tumor-induced autophagy [138]. Interestingly, lymphopenia has been associated with mitochondrial dysfunction [139]. In SARS-CoV-2 infections, mitochondria play an essential role as, aside from the interaction with SARS-CoV-2 NSPs, the virus can disrupt immune responses by altering the release of type I interferon (IFN-I) via mitochondria anti-viral signaling (MAVS) adaptor molecules. Indeed, viral proteins can activate caspases 8 and 9, promoting the release of cytochrome c, activating the mitochondrial death pathway. In addition, the SARS-CoV-2 virus can also cause gastrointestinal infections, altering the mitochondria–gut microbiota crosstalk, a key process for intestinal homeostasis [140].

## 7. Vagus Nerve Stimulation

In this section, we examine the physiological role of the vagus nerve and CAP in averting microbial translocation, highlighting the manipulation of cholinergic signaling as a treatment strategy for long COVID, including the “brain fog”.

The vagus nerve is composed of 20% efferent and 80% afferent fibers that support two-way communication between the brain and the body’s periphery. This is significant as it indicates that despite autonomy (ENS can manage the GI function after vagotomy), the gut is regulated centrally via the efferent vagus and neuro-immune pathways.

Novel studies have shown that the SARS-CoV-2 virus disrupts CAP, altering cholinergic signaling and the gut–brain axis [141]. In this regard, SARS-CoV-2 may exacerbate myasthenia gravis, emphasizing interference with neuronal and non-neuronal ACh [142,143].

Novel studies have found that α7nAChR dysfunction likely precedes Tau pathology, emphasizing the importance of cholinergic signaling in preventing tauopathies [144]. On the other hand, noninvasive taVNS may restore not only the integrity of the gut barrier but also “drain” viral reservoirs in microglia, monocytes, ECs, or IECs, eradicating the latent infection.

Currently, migraine headaches and some seizures are being treated with transcutaneous vagal nerve stimulation (tVNS), a well-tolerated modality that is easy to administer. A tVNS variant, spleen ultrasound neuromodulation, is based on the recent finding that memory T cells generate ACh in the gut and spleen, indicating that this modality could restore the homeostasis of adaptive immunity [145,146,147,148].

Taken together, the vagus nerve regulates ENS via CAP. At the molecular level, CAP influences intestinal permeability and gut microbiota composition via the AhR/STAT3/IL-22 axis. Under pathological conditions, this axis can be impaired, disrupting the gut–brain communication—a deficit that is restored by taVNS.

Recent studies have found that the SARS-CoV-2 virus can hijack the signal transducer and activator of transcription 1 (STAT1), hyperactivating STAT3 and disrupting the AhR/STAT3/IL-22 axis [149,150]. It was hypothesized that activated STAT3 maintains the symptoms of long COVID by altering immunity, a condition reminiscent of the rare genetic disorder STAT3 gain of function (GOF) [151,152,153]. Along this line, several studies have shown that α7nAChR stimulation inactivates STAT3, emphasizing the mechanism by which taVNS may ameliorate long COVID [154,155,156] (Figure 7). Indeed, taVNS lowers endothelial senescence, deactivates macrophages/monocytes/microglia, and restores gut-barrier homeostasis.

## 8. Conclusions

Long COVID is the proof of concept that the SARS-CoV-2 virus can linger in viral reservoirs, including microglia, EC, or IECs, engendering long COVID.

The S antigen of the SARS-CoV-2 virus contains fusogens that pierce host plasma membranes, initiating cell–cell fusion. Fused, aneuploid, or polyploid neurons, encountered in normal aging as well as in major cognitive disorders, likely account for the “brain fog” experienced by many patients with long COVID. Moreover, SARS-CoV-2 and HERVs “seed” pTau, suggesting that long COVID shares many pathogenetic traits with tauopathies. Furthermore, excessive pTau disrupts CAP and ACh signaling, compromising the gut barrier and contributing to the translocation of LPS and microbes outside the GI tract. LPS enhances viral infectivity and at the same time augments the generation of pTau, further linking long COVID to tauopathies.

taVNS restores the integrity of CAP as well as the gut-barrier function. In addition, upregulated intracellular ACh, a neurotransmitter with antiviral properties, may eradicate the virus from reservoirs, indicating its likely usefulness for both long COVID and HIV.

## Figures and Tables

**Figure 1 ijms-24-12648-f001:**
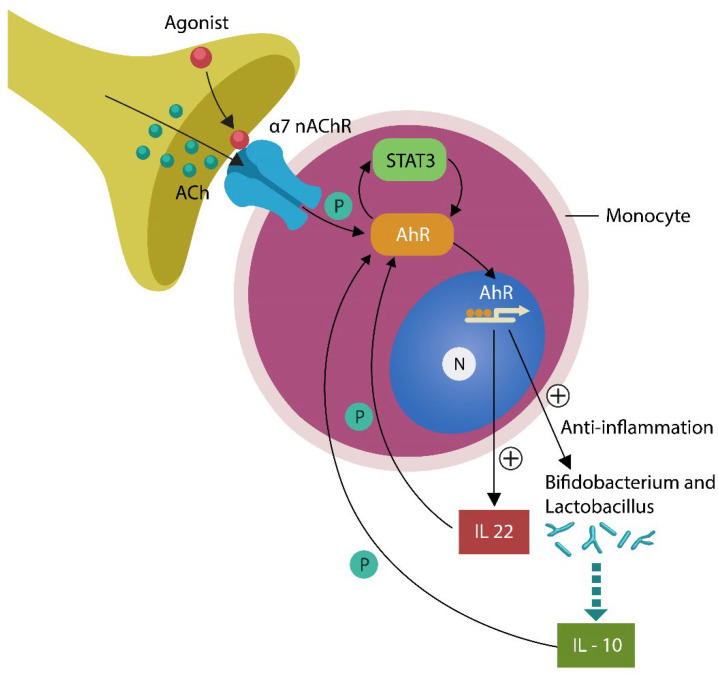
Monocytes are directly innervated by the vagus nerve (VN) via alpha 7 nicotinic acetylcholine receptor (α7nAChR). Impaired ACh signaling alters the AhR/STAT3/IL-22 axis, generating a SARS-CoV-2-friendly microenvironment. ACh phosphorylates AhR and STAT3, lowering viral infection. AhR enters the nucleus, where it facilitates the transcription of IL-22, promoting the beneficial gut microbes *Bifidobacterium* and *Lactobacillus*. These probiotics, in turn, phosphorylate AhR further. Aberrant phosphorylation of the AhR/STAT3/IL-22 axis promotes viral infection, disrupting gut-barrier homeostasis.

**Figure 2 ijms-24-12648-f002:**
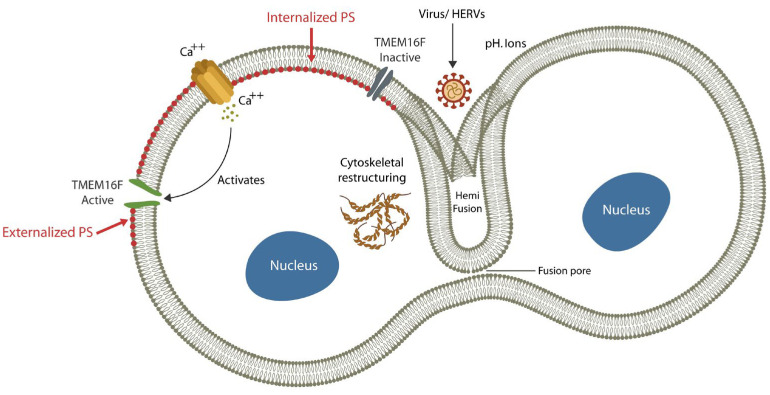
Exogenous and endogenous viruses can induce cellular senescence and syncytia formation by fusing host cells with each other. The molecular machinery of cell–cell fusion consists of Ca^2+^, TMEM16F, and externalized phosphatidylserine (ePS). Fusion is initiated by a fusion pore that gets larger until the cells share the cytoplasm, nuclei, and organelles. Local pH plays a major role in the initial step of pore formation.

**Figure 3 ijms-24-12648-f003:**
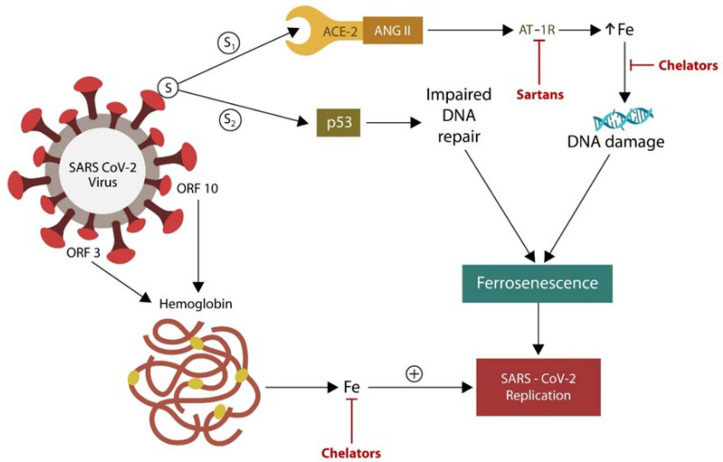
The SARS-CoV-2 virus requires iron for replication. It can obtain this biometal directly from hemoglobin via open reading frame 3 and 10 (ORF3 and ORF10) or indirectly by inducing ferrosenescence. The S_1_ antigen blockade of ACE-2 upregulates ANG II, which increases intracellular iron by overstimulating angiotensin II type 1 receptors (AT-1Rs). The S_2_ antigen of SARS-CoV-2 disables p53, disrupting DNA repair.

**Figure 4 ijms-24-12648-f004:**
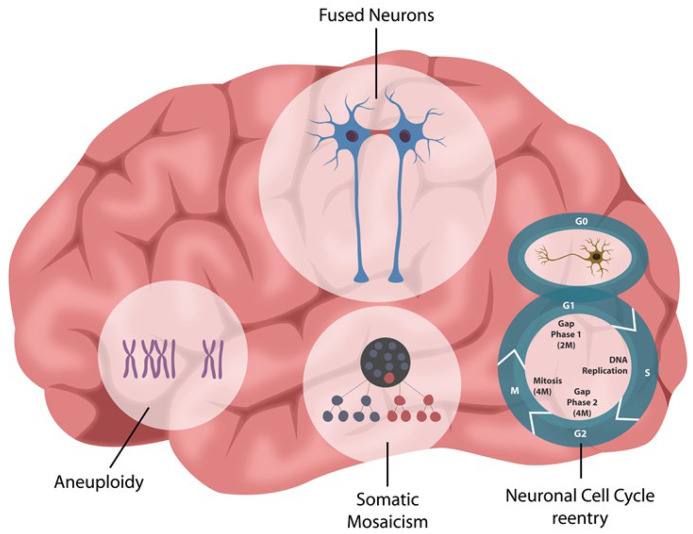
Cell–cell fusion in long COVID. The fact that mature neurons re-entering the cell cycle has been known for over two decades, however, the fact that this may be a consequence of viral infections has not been considered until the COVID-19 pandemic. Mature neurons engaging in mitosis can contribute to aneuploidy, polyploidy, and somatic mosaicism, documented by previous studies. Impaired information processing in fused neurons may explain COVID-associated neurocognitive dysfunction (CAND).

**Figure 5 ijms-24-12648-f005:**
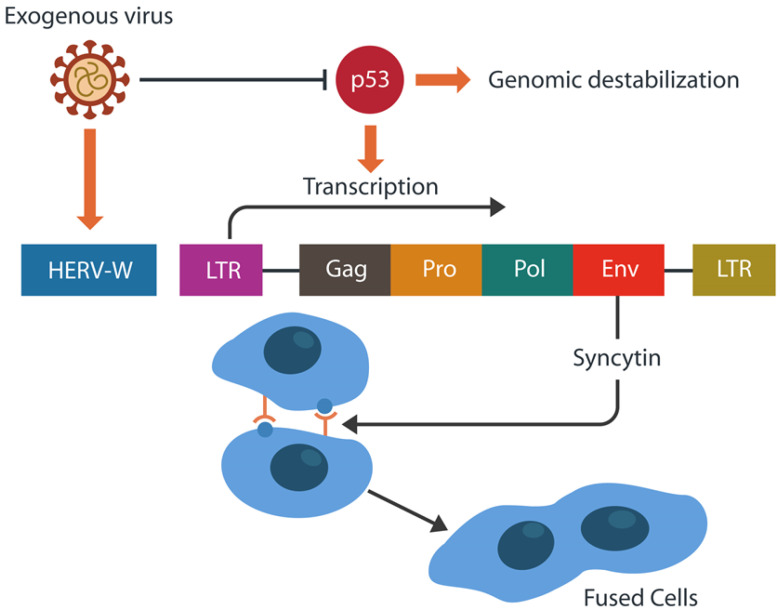
Exogenous viruses, including SARS-CoV-2, can activate human endogenous retroviruses (HERVs) and inhibit p53, destabilizing the genome. This activates the transcription of HERV-W ENV, resulting in excessive Syncytn-1, which promotes cell–cell fusion, including neuronal fusion.

**Figure 6 ijms-24-12648-f006:**
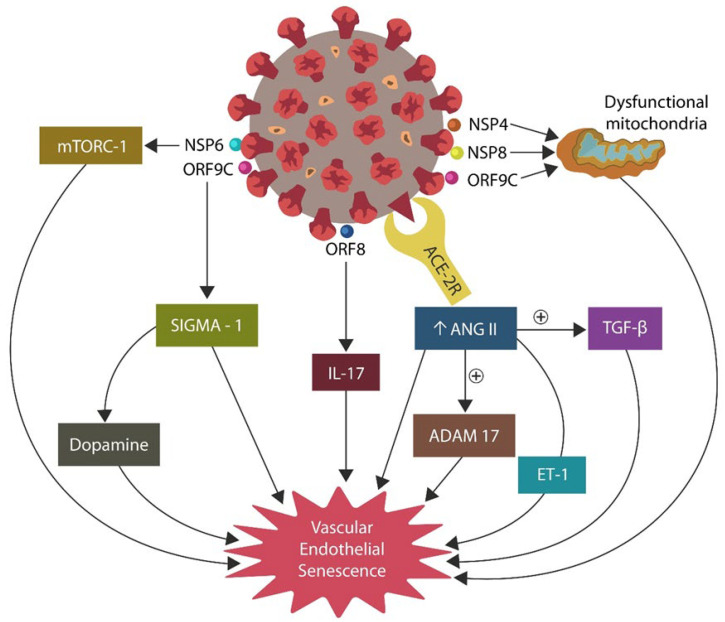
SARS-CoV-2/host protein–protein interactions, inducing endothelial senescence. Direct mechanisms include NSP6/mTORC1 and indirect ones include mitochondria, angiotensin II (ANG II), IL-17, and SIGMA-1/dopamine. Senescence activates endothelial cells (ECs) and alters the permeability of the gut barrier, enabling microbial migration outside the GI tract, a pathology likely accounting for the symptoms of long COVID.

**Figure 7 ijms-24-12648-f007:**
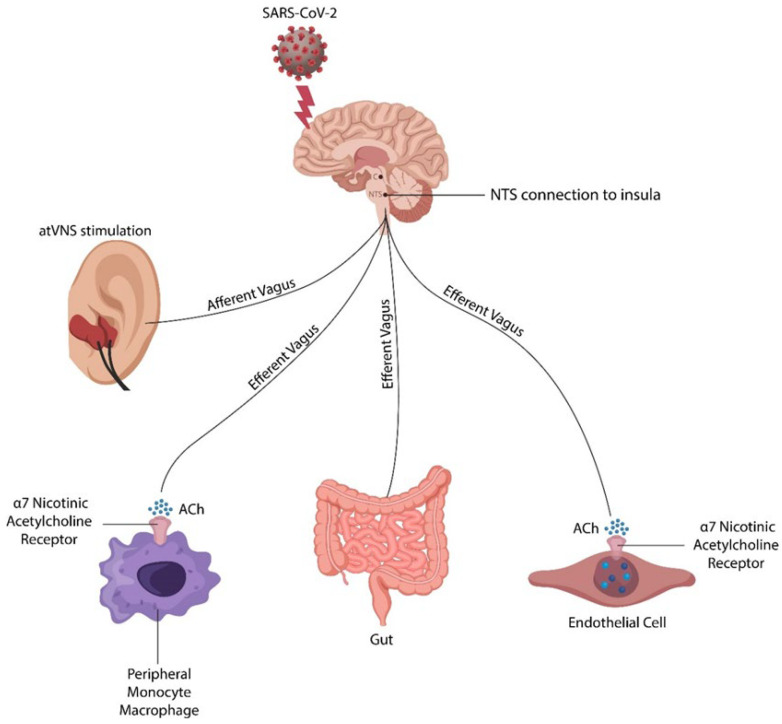
taVNS activates the auricular vagus nerve, opposing the long COVID-mediated “brain fog”. The efferent vagal fibers originate in the nucleus tractus solitarius and innervate microglia/monocytes/macrophages, ECs, and IECs via α7nAChR. Upregulated ACh exerts antiviral and anti-inflammatory properties, likely clearing the virus from reservoirs.

## Data Availability

Not applicable.

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
