# Peer review of "Long COVID as a Tauopathy: Of “Brain Fog” and “Fusogen Storms”"

_ijms, 2023, doi:10.3390/ijms241612648_

Round 1
Reviewer 1 Report
In this review the authors revised the current literature about neurotropic viruses, with the main focus of providing a whole picture of long COVID pathological implications.
The overall discussion of the topic is exhaustive and well-written. The cited bibliography is quite extensive and relevant. The reading is quite easy and pleasant, despite the very dense writing.
I’ve just some minor observations.
-
the citation list should be numbered… I had a lot of difficulties in matching the numbered citations in the text with the correspondant articles in the list…
-
As the term “tauopathies” is anticipated in the title, it would be useful to explain what they are from the beginning, to help the unexpert reader, Please consider to reshuffle the introduction (if possible), by placing the description of tauopathies at the beginning of the paragraph, to provide a better frame of the topic
-
Please consider to dedicate a specific section to the role of mitochondria in the SARS-CoV-2 infections adn the consequences on their omeostasis. Besides the interaction between SARS-CoV-2 non-structural proteins (NsSP) with mitochondria, SARS-CoV-2 infection is also able to disorganize the immune response (sometimes eliciting a “storm” of proinflammatory cytokines) by impairing and delaying type I interferon (IFN-I) response through the interaction of CoV accessory proteins with mitochondria anti-viral signaling (MAVS) adaptor molecules, e.g., SARS-CoV ORF3b and ORF9b. SARS-CoV proteins can activate caspases 8 and 9, determining cytochrome c release from the mitochondria and modulating the mitochondrial death pathway. In addition to direct interaction with mitochondria, SARS-CoV-2 is able to impair the homeostasis of these organelles by altering, via gastrointestinal infection, the crosstalk between mitochondria and the gut microbiota
-
Although the role of gut bacteria and bacterial metabolites are largely discussed, the concept of “gut-brain axis” is never clearly stated, with tis term.
-
Among the vast literature cited, I would like to suggest a more recent pubblication of the group of prof. Hilliard: doi: 10.1073/pnas.1919063117. Epub 2020 Aug 27.
-
line85: please explain the NSP acronymous (non structural protein)

Reviewer 2 Report
1.Please describe the complete form of the words before using the abbreviation of that words such as HIV, SARS-CoV-2. Please correct it.
2. Please add suitable key words under the abstract section for the submitted manuscript.
3. The authors should add the specific task for the manuscript preparation, The authors did not add anything and submitted according to the templates.
4. Please remove Institutional Review Board statement if it is not necessary to describe as well as for the informed consent statement.
5. Did the authors not get funding for the study? If the authors receive, please add at the funding section.
6.In abstract, the author said that they are looking for a potential marker but they did not describe anything in abstract. It should include in abstract.
7. In some figures, the authors used abbreviations and it should describe complete word as footnotes.
Reviewer 3 Report
This paper is an updated and comprehensive review focused on the hypothesis that long-covid manifestations share similar etiopathogenesis with tauopathies. A large amount of data is reviewed and linked together to support the hypothesis.
As clinician, I have some concern about the equivalence of "brain fog" with cognitive disorders. "Brain fog" is often ascertained in papers dealing with post-covid on the basis of patient self-report, without any objective evaluation, and has probably many components, including psychological distress. As clinician, I would appreciate very much a short paragaph better focusing on how the fused neurons could account for "brain fog" or cognitive disorders.
